

# Demystifying diagnosis: an efficient deep learning technique with explainable AI to improve breast cancer detection

Ahmed Alzahrani[1], Muhammad Ali Raza[2] and Muhammad Zubair Asghar[2]

[1] Department of Computer Science, Faculty of Computing and Information Technology, King Abdulaziz University, Jeddah, Saudi Arabia
[2] Gomal Research Institute of Computing, Faculty of Computing, Gomal University, D.I. Khan, KP, Pakistan

## ABSTRACT

As per a WHO survey conducted in 2023, more than 2.3 million breast cancer (BC) cases are reported every year. In nearly 95% of countries, the second leading cause of death for females is BC. Breast and cervical cancers cause 80% of reported deaths in middle-income countries. Early detection of breast cancer can help patients better manage their condition and increase their chances of survival. However, traditional AI models frequently conceal their decision-making processes and are mainly tailored for classification tasks. Our approach combines composite deep learning techniques with explainable artificial intelligence (XAI) to enhance interpretability and predictive accuracy. By utilizing XAI to examine features and provide insights into its classifications, the model clarifies the rationale behind its decisions, resulting in an understanding of concealed patterns linked to breast cancer detection. The XAI strengthens practitioners' and health researchers' confidence and understanding of artificial intelligence (AI)-based models. In this work, we introduce a hybrid deep learning bi-directional long short-term memory-convolutional neural network (BiLSTM-CNN) model to identify breast cancer using patient data effectively. We first balanced the dataset before using the BiLSTM-CNN model. The hybrid deep learning (DL) model presented here performed well in comparison to other studies, with 0.993 accuracy, precision 0.99, recall 0.99, and F1-score 0.99.

## INTRODUCTION

Breast cancer (BC) is one of the most common cancers affecting women worldwide and has become a leading cause of death. Breast cancer kills more than 40,000 women and nearly 600 men each year, according to the latest data from the American Cancer Society. There are four main types of breast cancer: benign, typical, localized, and invasive. Benign tumors cause minor changes in the anatomy of the breast, are not malignant, and are not as dangerous as malignant cancer (*Nasser & Yusof, 2023*).

Detecting breast cancer, in its stages poses a challenge due to its high fatality rate among women often leading to death. Identifying tumors at a stage can improve the likelihood of

Corresponding author
Muhammad Zubair Asghar,
mzubairgu@gmail.com

survival. Each year millions of women are diagnosed with breast cancer and many lose their lives to this illness; thus timely detection of tumors is crucial, for combating the disease and enhancing patient survival ratio. Furthermore, the basic instruments used to diagnose breast cancer are too expensive for low-income countries. Thus, reducing the physical and social impact of breast cancer on patients, is largely dependent on early detection (*Raza et al., 2024*).

Early detection of malignant cancer will assist combat the illness and increase the patient's chances of survival. The healthcare industry generates vast amounts of data daily, which can be analyzed using data mining techniques to uncover patterns and enable accurate clinical predictions. Even though, traditional machine learning systems have their benefits, they often operate like arcane contraptions with little understanding of the decision-making process, which undermines confidence in and understanding of their predictions (*Sengar, Gaikwad & Nagdive, 2020*; *Arshad, 2023*).

The aim of this study is to address these concerns by developing an explainable artificial intelligence (XAI) system that combines various methods to differentiate between cancerous and non-cancerous growths, in the body. This study integrates explainable AI techniques into a hybrid bi-directional long short-term memory-convolutional neural network (BiLSTM-CNN) model to improve diagnostic accuracy while providing clear and actionable insights into the model's predictions, enhancing its utility for healthcare professionals. The research aims to improve artificial intelligence (AI) technology in this field to create a user tool that can help detect and intervene at a stage to enhance the diagnosis of breast cancers effectively.

## Research motivation

The motivation, behind this research is driven by the requirement for timely identification of breast cancer. A major contributor to female mortality globally despite medical advancements in technology; diagnostic instruments frequently fall short in accuracy and cost effectiveness particularly in resource constrained environments. Additionally, various neural network models function as entities posing challenges for healthcare providers to have confidence, in their prognostications.

The study aims to enhance diagnostic precision and interpretability by integrating XAI into the hybrid BiLSTM-CNN model. This integration facilitates trust and usability for healthcare professionals, ultimately leading to improved patient outcomes. Moreover, the emphasis on effectiveness guarantees that this technique can be utilized in scenarios making cutting edge diagnostic resources more reachable and cost effective, for healthcare facilities globally.

## Problem formulation and research objectives

The objective of the work is to develop a BC predication system from given BC that can discriminate between malignant and benign cancer. The goal is to predict BC based on two class labels $Si \in \{malignant, benign\}$ *provided* $D = \{d1, d2 \ldots dn\}$ as a dataset. The XAI module will offer easily understandable rationales, for its predictions. This study aims to focus on the following main goals:

**RO1:** To apply hybrid deep learning (BILSTM+CNN) technique and explainable AI to diagnose breast cancer

**RO2:** To compare the proposed hybrid deep learning technique (BiLSTM-CNN) with various machine learning and deep learning models for breast cancer detection.

**RO3:** To assess the efficiency of the proposed model using ablation study.

## Research contributions

The key contributions of this research are as follows:

- Development of a hybrid BiLSTM-CNN deep learning model for breast cancer detection, offering improved diagnostic accuracy.
- Integration of XAI techniques, specifically Shapley Additive Explanations (SHAP), to enhance model interpretability and provide actionable insights for clinicians.
- Comprehensive evaluation of the proposed model's performance using rigorous benchmarking, ablation studies, and comparison with traditional machine learning and standalone deep learning models.
- Demonstration of the effectiveness of feature selection and data balancing techniques to improve the model's robustness and generalization.

The structure of rest of the paperwork is as follows: The literature review portion provides an overview of relevant material, followed by a discussion of the methodology, findings, and conclusions. The last section concludes with a description of future studies. Table 1 shows the glossary of key terms.

## Literature review

This segment includes an overview of the literature on disease diagnosis. *Sengar, Gaikwad & Nagdive (2020)* used the Waikato Environment for Knowledge Analysis (Weka) platform to study breast cancer. WBC dataset is analyzed using a machine learning (ML) decision tree classifier for the construction of an ontological model. The objective of this study was to classify malignant and benign tumors using a decision tree based on the SWRL and implement the ontological reasoner based on the decision tree. 97.10% accuracy was achieved by the ontological model (*Wang, 2022*). This article proposes a hybrid model (artificial neural network-support vector machine (ANN-SVM)) that combines artificial neural networks (ANNs) and support vector machines (SVMs) for the finding of BC. SVM is used as a BC classification algorithm and ANN as a feature extractor. Fine-tuning the SVM hyperparameters and adjusting the ANN structure might significantly enhance the intended model (*Mangukiya, Vaghani & Savani, 2022*). In this study, numerous ML approches for detecting BC were evaluated using the Breast-Cancer WBC dataset. In this research, BC is diagnosed using ANN, k-neural networks (K-NN), and SVM. *Massari et al. (2023)* performed this breast cancer prediction study using varieties of ensemble ML models on the WDBC dataset. Breast cancer is classified using different machine learning classifiers following data preparation and various feature selection algorithms. MLP and J48 approaches attained maximum accurateness using the

**Table 1 Glossary of key terms.**

| Term | Definition |
|---|---|
| Explainable AI (XAI) | Techniques that make AI models interpretable, providing insights into how predictions are made. |
| BiLSTM (Bidirectional Long Short-Term Memory) | A deep learning model capturing both past and future dependencies in sequential data for better context understanding. |
| CNN (Convolutional Neural Network) | A neural network architecture specialized in feature extraction and pattern recognition, commonly used for image and structured data. |
| Feature Selection (FS) | The process of identifying and selecting the most relevant features to improve model performance and interpretability. |
| SHAP (Shapley Additive Explanations) | An XAI technique based on game theory, assigning importance scores to features to explain model predictions. |
| AUC-ROC (Area Under the Receiver Operating Characteristic Curve) | A metric measuring a model's ability to distinguish between positive and negative cases across various thresholds. |
| Data Balancing | Techniques to address class imbalances by ensuring equal representation of all classes in the dataset. |
| Precision | The proportion of true positive predictions out of all positive predictions made by the model. |
| Recall (Sensitivity) | The proportion of true positive predictions out of all actual positive instances in the dataset. |
| F1-Score | The harmonic mean of precision and recall, balancing their contributions to evaluate model performance. |
| Receiver Operating Characteristic (ROC) Curve | A graphical representation of a model's sensitivity *vs.* specificity, used to evaluate classification performance. |
| Correlation Coefficient (r) | A statistical measure that indicates the strength and direction of a linear relationship between two variables. |
| Deep Learning | A subset of machine learning that uses neural networks with multiple layers to learn patterns in data. |
| Sequential Data | Data where the order of elements is significant, such as time series or text sequences. |
| Class Imbalance | A condition in datasets where some classes have significantly more samples than others, often leading to biased models. |

genetic search feature selection approach. *Pang (2022)* investigated the performance of ANNs and decision trees (DTs) for BC prediction. For the construction of the DT model, various preprocessing techniques and PCA are used. DT (PCA transformation) is the most accurate with 96% accuracy (*Hall, Chang & Mitchell, 2022*). There are thousands of females who die every year from BC. The DL model is used in this article to diagnose breast cancer. Additionally, ML algorithms are analyzed in terms of their number. In this research, CNN's deep learning model is primarily examined. In order to improve accuracy, some other deep learning algorithms can be adopted. In this work, *Lopez-Martin et al. (2017)* describes XAI's function in the medical field. They employ several XAI methods, such as LIME and SHAP, in addition to applying machine learning models to create counterfactuals and anchors using the CVD dataset. This study highlights the necessity of openness and understandability in AI-based models, particularly when making decisions that affect human life, such as in medical diagnosis. The black box and understandability decision-making processes are explained by this research. In this study, the deep learning model may also be utilized to improve the understandability and transparency of the AI-based model, even though just the machine learning model is utilized to grasp XAI.

*Solanki et al. (2021)* present a ML approach for BC classification and use XAI technique (SHAP) to emphasizing inheritability. Using K-NN, SVM, XG-Boost, random forest (RF), and ANN to achieving high accuracy and precision in categorizing BC disease. Key attributes like "bare nuclei" and "area worst" are highlighted as important for BC prediction. In this research the K-NN ML model outperformed with accuracy of 97.7%. In recent years, XAI has gained significant attention in healthcare for improving the interpretability and trustworthiness of diagnostic models. Several studies have introduced innovative XAI methodologies. *Aziz et al. (2024)* conducted a comprehensive review of XAI techniques in clinical decision support systems, emphasizing their role in improving transparency and aiding clinicians in interpreting model predictions. *Demir et al. (2024)* proposed transformer-based prototypes for medical diagnoses, leveraging explainability to enhance user trust and diagnostic accuracy, particularly in breast cancer detection. *Munshi et al. (2024)* explored ensemble-based XAI models for breast cancer detection, achieving a significant improvement in interpretability and accuracy. *Islam et al. (2024)* presents a comprehensive XAI evaluation framework incorporating fidelity, interpretability, robustness, fairness, and completeness into a dynamic and adaptable scoring system. This framework evaluates various XAI methods, including SHAP and Grad-CAM, across multiple domains, including healthcare. While it does not specifically integrate SHAP and Grad-CAM into a single hybrid method, it provides valuable insights into the effectiveness of these techniques in medical applications.

### Research gap

While deep learning models aid breast cancer diagnosis, many studies struggle with classification performance due to suboptimal architecture and parameter choices. This work introduces a BiLSTM-CNN hybrid model with SHAP-based explainability to enhance accuracy and interpretability. By optimizing configurations and benchmarking against prior models, the study improves diagnostic reliability and trust in AI-driven predictions.

## MATERIALS AND METHODS

In the proposed methodology, there are three main components: dataset acquisition, data pre-processing, and an overview of the proposed model (See Fig. 1). Each of the submodules is described in more detail below:

### Dataset acquisition

In this study, UCI's repository was used to access Wisconsin Breast Cancer Diagnostic (WDBC) data (*UCI Machine Learning Repository, 2019*) available publically at: https://archive.ics.uci.edu/ml/datasets/breast+cancer+wisconsin+%28diagnostic%29. There are 569 instances in the data set overall, and each instance has 32 properties. A total of 357 of the 569 cases are benign, while 212 are malignant. The dataset there has no missing record. Details of the given WDBC dataset are presented in Fig. 2.

A fine needle aspiration (FNA) is used to create the characteristics of the digital images. One type of biopsy procedure is called FNA, where samples are taken by inserting a tiny,
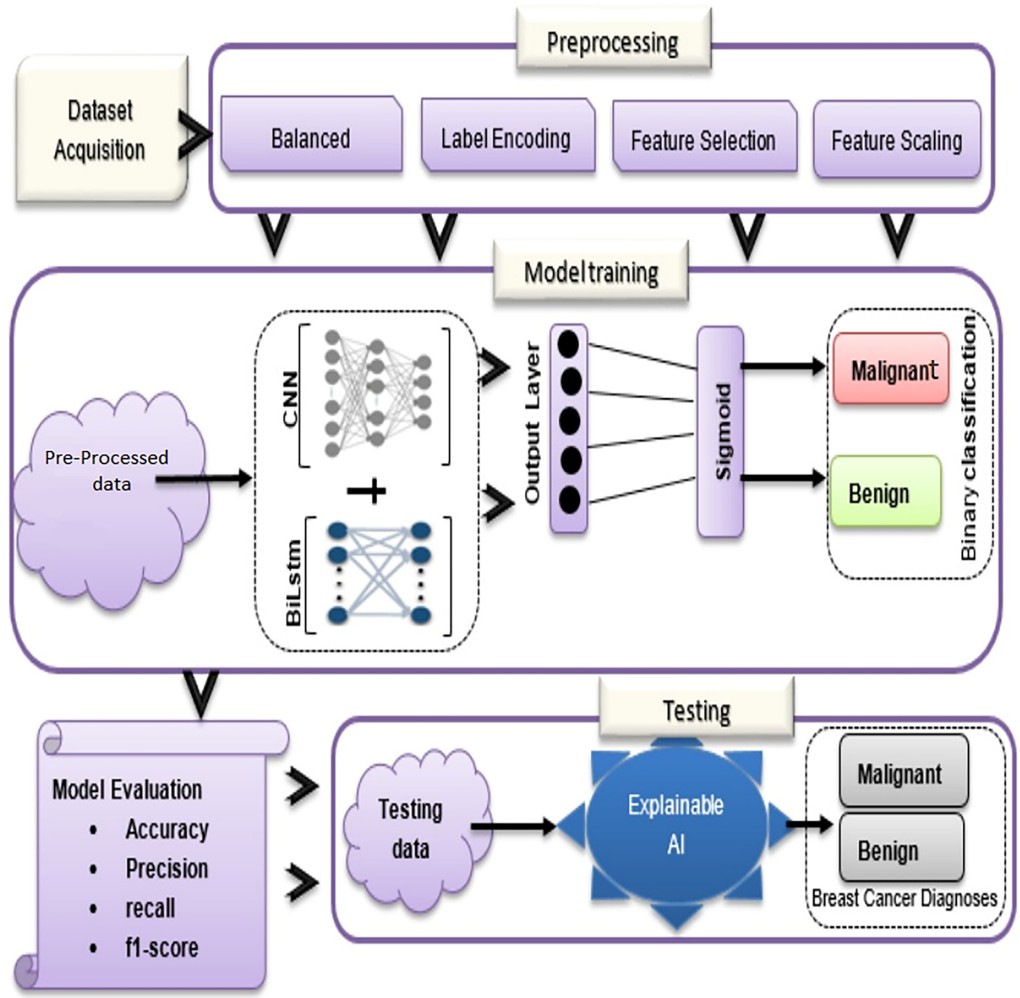

**Figure 1 Overview of BiLSTM-CNN methodology with explainable AI.**

pointed needle into a region of bodily fluid or tissue that seems abnormal. In the images the attributes/characteristics of the visible nuclei determine input features (*Darya, Nassif & Al-Shabi, 2022*). In total, thirty attributes were computed for each image, including the worst, highest, mean, and standard error (SE). The values of all features are recorded using four significant digits (*Abdulla, Sagheer & Veisi, 2021*).

### Data splitting strategy

The Wisconsin Breast Cancer Diagnostic dataset, while widely used and appropriate for breast cancer classification, is recognized as limited in diversity due to its reliance on a single data source. To mitigate potential biases and ensure robust evaluation, the dataset was divided into training, validation, and testing sets using a stratified sampling approach, ensuring balanced representation of malignant and benign cases (see Fig. 3).

| S.NO. | Features/Attribute Details |
|---|---|
| 1. | Radius (Mean of distances from center to points on the perimeter) |
| 2. | Texture (Standard deviation of gray-scale values) |
| 3. | Perimeter |
| 4. | Area |
| 5. | Smoothness (Local variation in radius lengths) |
| 6. | Compactness (Perimeter^2/area - 1.0) |
| 7. | Concavity (Severity of concave portions of the contour) |
| 8. | Concave-points (Number of concave portions of the contour) |
| 9. | Symmetry |
| 10. | Fractal-dimension ("coastline approximation" - 1) |

**Figure 2  An overview of the WDBC dataset.**     

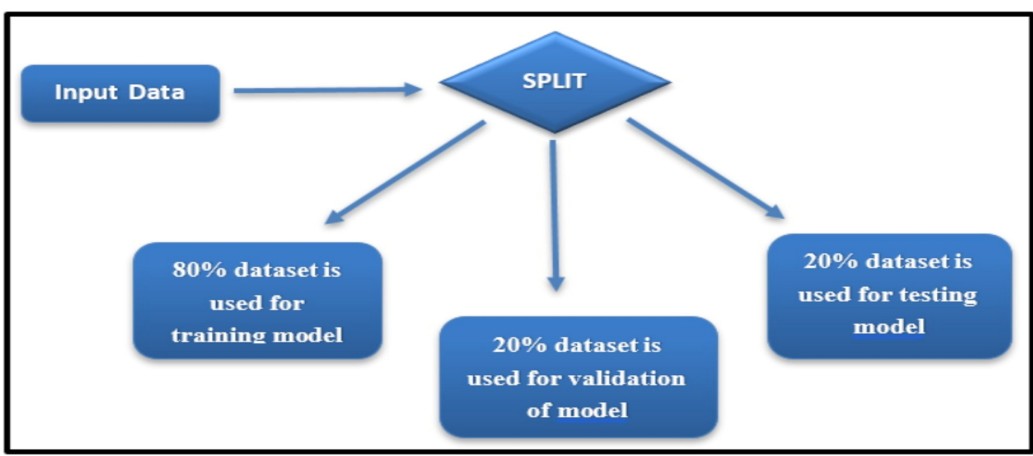

**Figure 3  Dataset split for training, validation, and testing.**

Stratification ensured that the proportion of malignant and benign cases in each subset mirrored the overall dataset distribution, thereby preserving the class balance during model training and evaluation.

The dataset was divided into three subsets:

- **Training set:** 80% of the data, used for model training.
- **Validation set:** 20% of the training set, employed for hyperparameter tuning.
- **Testing set:** 20% of the data, reserved for evaluating the final model's performance.

This approach helps maintain a consistent class distribution and reduces the risk of overfitting or bias during evaluation. Future studies will address the dataset's limitations by incorporating additional publicly available breast cancer datasets and performing external validation to improve generalizability across diverse populations.

### Treatment of data

Cross fold validation (CFV) is used to validate the model. Subset of the training instance are created and stored at each level. The basis for our investigation was a different "holdout" approach. The sample with greatest F1 score was selected as the holdout sample.

## Pre-processing

The first and most important stage in getting the best categorization results is pre-processing. Before classifying the data, it is often applied to ensure the intended results are produced.

### Feature selection

Feature selection (FS) has confirmed to be an effective and efficient data preparation method for a range of DM and ML tasks. This is especially true for multi-dimensional data. The primary objectives of the feature selection procedure are to provide data that is easily comprehensible and to enhance data mining performance (*Jasim et al., 2022*).

In this study, the Extra Trees Classifier (ETC) feature selection approach was used.

## Rationale for choosing the extra trees classifier

FS is critical for improving the interpretability and efficiency of machine learning models. In this study, we employed the ETC for FS, which is an ensemble learning method known for its ability to handle high-dimensional data while preventing overfitting. The ETC ranks features based on their importance derived from decision-tree splits, providing both computational efficiency and interpretability.

Compared to alternatives such as principal component analysis (PCA) and Lasso, the ETC offers several advantages. PCA, while effective in dimensionality reduction, transforms features into uncorrelated components, which compromises interpretability—a key requirement in medical diagnostics. Lasso regression, though capable of feature selection, is sensitive to multicollinearity and may require extensive hyperparameter tuning to achieve optimal performance. By contrast, the ETC explicitly identifies the most informative features without such limitations.

The Extra Trees Classifier, sometimes called the Extremely Randomised Tree (ERT) Classifier, is a type of ensemble ML algorithm that generates classification outputs by aggregating the observations of several disconnect DT collected in a forest (*AlindGupta, 2023*). In a technique that is similar to random forest, Extra Trees Classifier jumbles up

certain decisions and subsets of data to prevent overfitting and overlearning from the data (*Li et al., 2018*).

In this instance, information gain (IG) and entropy are what determine which traits are the best (*Islam et al., 2024*). Below are the formulas for entropy and information gain.

$$\text{Entropy}(E) = \sum_{i=1}^{z} -h_i \log_2(h_i) \tag{1}$$

$$\text{IG}(E, G) = E - \sum_{c \in \text{Values}(G)} \frac{|Ec|}{E} Ec \tag{2}$$

where $z$ = number of uniquely labeled output classes, $h_i$ = "the proportion of rows with an output label is $i$" and entropy (E) = "training set".

An ETC model was built after splitting the data set between 0.80 and 0.20 in order to evaluate the significance of each feature. We selected a subset of 18 variables to develop our learning model from the breast cancer diagnostic dataset. Table 2 presents the top 18 attributes selected using ETC, prioritized based on their relevance to the output feature. Attributes with the highest scores were given preference, as they exhibited a strong correlation and dependency on the output class. These features were utilized for training the BiLSTM-CNN model, playing a crucial role in enhancing its diagnostic performance.

Table 2 provides useful insights into feature importance and correlation but raises several critical concerns. The top features, such as perimeter_worst and area_worst, dominate the model, suggesting potential over-reliance on a limited subset. Meanwhile, the marginal contributions of other features question their practical relevance. While high correlations reflect strong linear relationships, they overlook nonlinear dependencies and interactions that advanced models might exploit. Additionally, redundancy among features, such as radius_mean and area_mean, may introduce multicollinearity, impairing model interpretability.

### Unbalanced dataset management

To address the class imbalance in the WDBC dataset (357 benign and 212 malignant cases), we applied random oversampling to equalize the number of samples in each class (*Das et al., 2023*). The minority class (malignant) was oversampled to match the majority class (benign), resulting in a balanced dataset of 714 instances (357 per class).

To ensure that oversampling did not introduce bias or overfitting:

1) **Cross-validation**: We employed 10-fold cross-validation, where the dataset was split into training and testing folds at each iteration. Oversampling was performed exclusively on the training folds to prevent data leakage into the testing folds, ensuring an unbiased evaluation of the model's performance.
2) **Performance metrics**: Metrics such as accuracy, precision, recall, and F1-score were monitored to assess the model's generalization. High recall and F1-scores for the minority class indicated that the oversampling approach improved classification without compromising model reliability.

**Table 2 Top 18 features selected for optimal classification.**

| Feature | Importance (ETC) | Correlation (r) |
|---|---|---|
| perimeter_worst | 0.152 | 0.84 |
| area_worst | 0.143 | 0.81 |
| radius_mean | 0.129 | 0.78 |
| texture_mean | 0.102 | 0.76 |
| smoothness_worst | 0.098 | 0.74 |
| compactness_mean | 0.089 | 0.73 |
| concavity_mean | 0.081 | 0.71 |
| concave points_worst | 0.075 | 0.69 |
| radius_se | 0.071 | 0.67 |
| area_mean | 0.068 | 0.65 |
| fractal_dimension_mean | 0.064 | 0.63 |
| symmetry_mean | 0.061 | 0.61 |
| compactness_worst | 0.058 | 0.59 |
| concavity_worst | 0.054 | 0.57 |
| symmetry_worst | 0.049 | 0.55 |
| texture_worst | 0.045 | 0.53 |
| fractal_dimension_worst | 0.041 | 0.51 |
| smoothness_mean | 0.037 | 0.49 |

3) **Regularization**: Dropout layers (rate 0.3) were incorporated in the BiLSTM-CNN architecture to prevent overfitting during training on the augmented dataset.

### Label encoding and feature scaling

There are 32 properties in the provided data set; the first one, id, adds nothing to our ability to determine if the data set is benign or malignant. As a result, the dataset's id column was removed. "Diagnosis" is one of the features in the dataset, which belongs to the character class. As it is widely recommended that DL models only receive numeric data, the label encoder in the scikit-learn package transforms this column into a numeric type (*Roy et al., 2021*).

A crucial aspect of many ML and DL models is ensuring that the data is scaled sensibly. Feature scaling is a standardization method ensure that the data free scaled by converting the statistical scattering of the data into the following format: There is zero mean. The one is standard deviation (SD) (*Asghar et al., 2021*).

Equation (3) was used for feature scaling to standardize the dataset:

$$zi = (si - \bar{s})/\sigma. \tag{3}$$

In each feature, the $(\bar{s})$ mean and $(\sigma)$ standard deviation will be calculated. The feature standardized values are obtained by deducting the mean from each observation and dividing the result by the standard deviation. Once the given data has been distributed into training and test sets, standardization is performed. To standardize the training and test

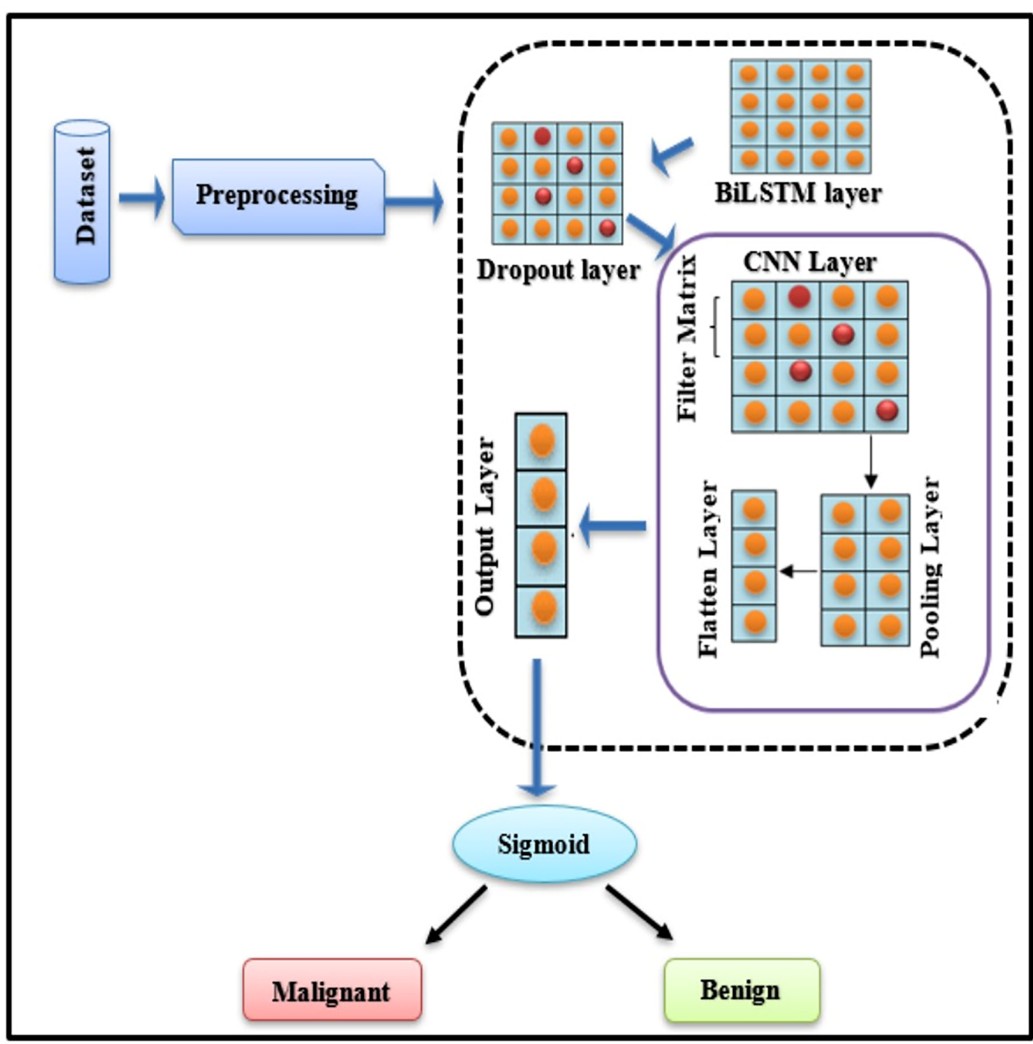

**Figure 4  BiLSTM-CNN hybrid model architecture.**

data sets, "StandardScalar().fit_transform()" and "StandardScalar().transform()" were used.

In the following section, data is supplied into a proposed approach (BiLSTM-CNN) after pre-processing. An explanation of the Bi-LSTM+CNN model for identifying BC is provided in this section.

## Breast cancer diagnoses using deep learning hybrid models

To diagnose BC, a BiLSTM with a CNN model is being established. When BC is detected, the technique categorises it as "Malignant" (cancerous tumour) or "Benign" (normal tumour). Fig. 4 explains the basic framework of the proposed study.

## Novelty and rationale for the BiLSTM-CNN architecture

The proposed BiLSTM-CNN hybrid architecture represents a novel approach for breast cancer diagnosis using tabular data, addressing limitations in existing methods. BiLSTM, a

bidirectional extension of LSTM, excels at capturing sequential patterns and temporal dependencies, which are crucial for extracting meaningful feature interactions in structured datasets. Complementing this, CNN efficiently identifies localized dependencies among features through convolution operations. This synergy enhances both diagnostic accuracy and interpretability, providing a richer data representation for subsequent analysis.

Unlike traditional machine learning models or standalone deep learning architectures (*e.g.*, CNN or LSTM), this hybrid approach uniquely integrates spatial and sequential feature modeling tailored for tabular data. This dual capability not only boosts predictive performance but also facilitates deeper insights into the data through explainability techniques such as SHAP (*Aziz et al., 2024*). The inclusion of CNN aids in hierarchical feature extraction, while BiLSTM ensures the preservation of contextual information across feature sequences, which is particularly beneficial for medical diagnostics. This novel combination has demonstrated superior performance in diagnostic accuracy and interpretability compared to benchmark studies, as evidenced by the results presented in this work.

## Comprehensive design of the BiLSTM-CNN model

The BiLSTM-CNN model was optimized by systematically testing various configurations during the experimental phase. The final architecture consisted of two bidirectional LSTM layers, followed by two convolutional layers, a max-pooling layer, a dropout layer, and a fully connected dense layer. The BiLSTM layers, designed to capture sequential dependencies, comprised 64 and 128 units, respectively, and employed the Tanh activation function. To prevent overfitting, a dropout rate of 0.3 was applied to both BiLSTM layers. The convolutional layers, tasked with feature extraction, were configured with 32 filters each and a kernel size of 7, with ReLU activation. The max-pooling layer utilized a pool size of 2×2 to reduce dimensionality while preserving essential feature information. Finally, a fully connected dense layer with a sigmoid activation function enabled binary classification, distinguishing between malignant and benign cases.

The model was trained for 50 epochs using a batch size of 32, with the Adam optimizer (learning rate of 0.001) employed for efficient gradient descent. Early stopping with a patience of five epochs was implemented to mitigate overfitting. The dataset was split into training, validation, and testing subsets, ensuring that 20% of the training data was reserved for validation. To evaluate the model's robustness, a 10-fold cross-validation approach was employed, with performance assessed using metrics such as accuracy, precision, recall, F1-score, and the area under the receiver operating characteristic curve (AUC-ROC). All experiments were conducted on an NVIDIA Tesla V100 GPU (32 GB VRAM) using TensorFlow/Keras (version 2.9.1) in a Python 3.8 environment.

### The Bi-LSTM layer

As a result of its ability to incorporate past and future context, as well as sequence information, the Bi-LSTM model has been receiving considerable attention in recent years. As a result, it prevail over the limitations of traditional DL model (RNN), which can only

hold data for a short while and on other hand LSTMs, which can only remember context from the past (*Mulani, 2022*).

The Bi-LSTM layer is composed of the forward and backward LSTM subnetworks. Given an input sequence of (x1, x2, x3,…, xn) words, both forward and backward hidden layer vectors are computed by the Bi-LSTM. Representations of the right and left sequences ($l = \overleftarrow{l}, \overrightarrow{l}$) are concatenated to form the output sequence (l1, l2, l3,…, lt). Afterward, it becomes an input for the upper layer, which generates predictions based on every input (*Alghazzawi et al., 2021*).

### Dropout layer

Using the drop-out layer keeps the neural network from overfitting. There is a range from 0 to 1 for the dropout parameter (rate). Depending on how the BiLSTM layer implements dropout, neuron activity is blocked or deleted randomly. In this system, convolutional layer extracts feature, max pooling layer reduces input dimensionality, and flatten layer converts pooling layer's output into feature vectors. In the following section, we will explain the CNN layers. But firstly, we discuss, why CNN has been chosen.

CNNs are traditionally used for spatial data due to their ability to capture local dependencies *via* convolution operations. However, their applicability extends to tabular datasets by treating features as structured inputs. In this study, CNNs are employed to capture interactions between features by using 1D convolutional layers. These layers, combined with max-pooling operations, extract hierarchical feature representations, reducing dimensionality and enhancing the learning process. When integrated with BiLSTM, CNNs provide enriched input sequences that improve the temporal context understanding of the hybrid model. To adapt CNNs for tabular data, specific modifications, such as 1D convolutions with tailored filter sizes and strides, were implemented to emphasize feature-level relationships.

### Convolutional layer

A convolution is an procedure that transforms two functions into a third by adding mathematical operations. Using this procedure, you'll need an input matrix $K \in R^{vxw}$, the filter matrix $P \in R^{ixj}$, and the output matrix (F), which is also referred to as a feature map (*Roy et al., 2021*; *Asghar et al., 2021*).

The following is the representation of the feature mapping F:

$$F_{nm} = R(G \circ s_{n:n+k-1,m+c-1} + b) \tag{4}$$

where "$\circ$" is the convolution operation between $K \in R^{vxw}$ and $P \in R^{ixj}$ where R denotes Activation Function ReLU, b is the bias vector and G is the weight matrix.

### Max-pooling layer

As a result of the convolutional layer, the input feature maps are downsampled, which are then turned into sub-sampled feature maps by the MaxPooling layer. There is a reduction in the dimensionality of the feature map without a corresponding reduction in information volume It also shortens computation times and minimises overfitting (*Mulani, 2022*).

Mathematically, the MaxPooling operation is written as:

$$M_{nm} = \max(F_{n+k-1,m+c-1}) \tag{5}$$

### Flatten layer

Max-pooling output is transformed into feature vectors by flattening. A vector is created from the features so that the last set of fully connected layers can be applied to it, concluding the modeling process. The flatten layer can be used as follows (*Ambreen et al., 2024*):

$$L = \text{Reshaped polled}[(x - s + 1) * (y - u + 1)] \tag{6}$$

### Output layer/classification layer

For the final layer, sigmoid function output and CNN output are used to estimate the likelihood of recognizing the target labels accurately. In order to find the result at the end, use Eq. (7):

$$C_i = \sum g_i v_i + b \tag{7}$$

where "G" is the weight vector, "b" is the bias, and "v" is the input vector. Equation (8) explains how to calculate the sigmoid function.

$$\text{Sigmoid}(C_i) = 1/1 + e^{-C_i} \tag{8}$$

## Explainable AI with SHAP

Shapley Additive Explanations (SHAP) improves diagnostic transparency by assigning importance scores to individual features influencing model predictions. This ensures that healthcare practitioners can interpret and trust the model's diagnostic decisions (*Alghazzawi et al., 2021*).

While multiple interpretability methods are available, SHAP was selected due to its theoretical robustness and practical advantages. SHAP, based on Shapley values, ensures consistency and additivity in feature importance explanations. This is particularly crucial for the hybrid BiLSTM-CNN model, where complex interactions between temporal and feature-based inputs require reliable interpretability. Unlike LIME, which depends on local data sampling and may produce inconsistent results, SHAP provides stable explanations across both global and local perspectives. Additionally, SHAP effectively captures feature dependencies, a limitation of permutation importance, making it more suitable for models with intricate architectures. This choice aligns with the study's emphasis on transparent and trustworthy predictions for medical diagnostics (*Alghazzawi et al., 2025*).

### SHAP theory

Based on this approach, a SHAP value is determined for each attribute in the deep learning model to determine how each attribute contributes to the target value. Based on the

conditional expected value determined by deep learning, SHAP values are determined for every feature. In cooperative game theory, a solution idea known as a Shapley value divides the total gain that is attained through collaboration among game participants according to the insignificant contributions of each player. Using the formula below, we can determine the SHAP value (*Alghazzawi et al., 2025*):

$$\phi i(f) = \sum_{S \subseteq \{1,...,M\} \setminus \{i\}} \frac{|S|!(M - |S| - 1)!}{M!} [f(S \cup \{i\}) - f(S)] \tag{9}$$

f(S) is the output of the model when considering only the features in S and f(S∪{i}) is the model's output when considering feature, I in addition to the features in S. Where M is the total number of features, S is a subset of features excluding feature I, and |S| indicates the cardinality of the subset S.

### SHAP integration for explainable breast cancer diagnosis

The hybrid BiLSTM+CNN model calculates SHAP values, for the temporal features (from BiLSTM) and spatial features (from CNN). This helps determine the factors that influenced the models decisions and makes it easier for clinicians to understand the models predictions. In breast cancer diagnosis, as an example; SHAP can point out regions in an image or critical moments, in a patients record that influenced whether a diagnosis was malignant or benign. To enhance interpretability, the SHAP framework was utilized to explain the predictions of the BiLSTM-CNN model. SHAP was implemented using the SHAP Python library (version 0.41.0), leveraging the KernelExplainer method for feature attribution. The explainer was initialized with the pre-trained BiLSTM-CNN model and a random subset of 100 training samples to compute Shapley values, thereby quantifying the contribution of each input feature to the model's predictions. The analysis focused on the 18 features selected *via* the Extra Tree Classifier, which were shown to have the highest relevance for breast cancer diagnosis.

SHAP values were calculated to provide both local and global interpretability. Local explanations were derived for individual predictions, highlighting the specific features influencing the classification of a given instance as malignant or benign. Global explanations were generated to identify the overall importance of features across the dataset. Visualizations, including summary plots, force plots, and decision plots, were produced using SHAP's visualization toolkit and Matplotlib (version 3.5.2). These visualizations provided actionable insights for clinicians by elucidating the rationale behind the model's diagnostic decisions.

To validate the reliability of the SHAP analysis, multiple iterations were performed to ensure consistency in feature importance rankings. Additionally, the SHAP results were cross-compared with alternative interpretability techniques, such as Local Interpretable Model-agnostic Explanations (LIME) and permutation importance, confirming the robustness of the feature attributions. The SHAP framework's consistency and theoretical

robustness make it particularly suitable for the hybrid BiLSTM-CNN architecture, which involves complex interactions between temporal and spatial features.

## Applied example

With the help of the available dataset, numerous calculations were carried out to predict breast cancer diagnoses. Based on Eq. (7), the offered DL hybrid model is well capable to explain the functions it performs.

In the case of Malignant, the class label is "C1".

$C1 = g_1 \times v_1 + g_2 \times v_2 + b$

$C1 = 0.9 \times 0.7 + 0.7 \times 0.6 + 0.5 = 1.55$

In the case of benign, the class label is "C2".

$C2 = g_1 \times v_1 + g_2 \times v_2 + b$

$C2 = 0.4 \times 0.2 + 0.3 \times 0.2 + 0.5 = 0.64$

With the sigmoid activation function (8), each label's likelihood is calculated as follows:

$\text{Sigmoid}(C_1) = 1/1 + e^{-Q1}$

$\text{Sigmoid}(C_1) = 1/1 + e^{-1.55} = 0.826$

In the same way, activation function (sigmoid) was calculated for the other second class:

$\text{Sigmoid}(C_2) = 1/1 + e^{-Q2}$

$\text{Sigmoid}(C_2) = 1/1 + e^{-0.64} = 0.657$

Based on this calculation, the C1-Malignant (breast cancer) had the highest probability. Based on this given data, we can predict that the chance of breast cancer is "C1" (Fig. 5).

The model's outputs for various feature combinations:

$f(\{F1\}) = 0.2, f(\{F2\}) = 0.4, f(\{F3\}) = 0.6, f(\{F4\}) = 0.3, f(\{F1, F2\}) = 0.5,$
$f(\{F1, F3\}) = 0.7, f(\{F2, F3\}) = 0.8, f(\{F1, F4\}) = 0.45, f(\{F2, F4\}) = 0.65,$
$f(\{F3, F4\}) = 0.75, f(\{F1, F2, F3\}) = 0.9, f(\{F1, F2, F4\}) = 0.7,$
$f(\{F1, F3, F4\}) = 0.85, f(\{F2, F3, F4\}) = 0.95.$

**Calculations:**

1. **SHAP Value for F1:** $\phi F1(f) = [(0.9 - 0.5) + (0.9 - 0.7) + (0.9 - 0.45) + (0.9 - 0.75) + (0.9 - 0.65) + (0.9 - 0.3)]/(2^4 - 2^3) * (0.2 + 0.4 + 0.6)\phi F1(f) = [0.4 + 0.2 + 0.45 + 0.15 + 0.25 + 0.6]/6 * 1.2\phi F1(f) = 2.1/6 * 1.2\phi F1(f) = 0.42$

2. **SHAP Value for F2:** $\phi F2(f) = [(0.9 - 0.5) + (0.9 - 0.8) + (0.9 - 0.45) + (0.9 - 0.65) + (0.9 - 0.3) + (0.9 - 0.75)] / 6 * 1.2 \ \phi F2(f) = [0.4 + 0.1 + 0.45 + 0.25 + 0.6 + 0.15] / 6 * 1.2 \ \phi F2(f) = 1.95 / 6 * 1.2 \ \phi F2(f) = 0.39$

3. **SHAP Value for F3:** $\phi F3(f) = [(0.9 - 0.7) + (0.9 - 0.8) + (0.9 - 0.45) + (0.9 - 0.65) + (0.9 - 0.3) + (0.9 - 0.5)] / 6 * 1.2 \ \phi F3(f) = [0.2 + 0.1 + 0.45 + 0.25 + 0.6 + 0.4] / 6 * 1.2 \ \phi F3(f) = 2.0 / 6 * 1.2 \ \phi F3(f) = 0.4$

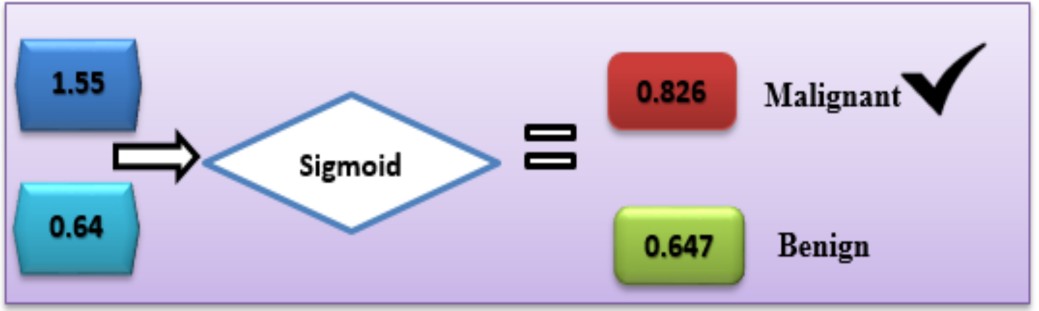

Figure 5 Use of the sigmoid function to diagnose breast cancer.

4. **SHAP Value for F4:** $\phi F4(f) = [(0.9 - 0.5) + (0.9 - 0.6) + (0.9 - 0.7) + (0.9 - 0.3) + (0.9 - 0.45) + (0.9 - 0.5)] / 6 * 1.2$ $\phi F4(f) = [0.4 + 0.3 + 0.2 + 0.6 + 0.45 + 0.4] / 6 * 1.2$ $\phi F4(f) = 2.35 / 6 * 1.2$ $\phi F4(f) = 0.47$

In this below example (Fig. 6) we can see how SHAP (SHapley Additive exPlanations) aids in understanding the predictions of a BiLSTM + CNN model, for diagnosing breast cancer cases. This tabulated information showcases how various features contribute to the models forecast for a scenario and categorizes the outcome as malignant or benign.

The SHAP values in Fig. 6 highlight key features influencing breast cancer diagnosis. Positive SHAP values indicate malignancy, while negative values suggest benignity. For malignant cases, tumor size (F1: 0.34, 0.35), shape (F2: 0.35), and irregular borders (F3: 0.37) contribute positively. Conversely, for benign cases, tumor size (−0.15) and borders (−0.05) have negative SHAP values, reducing malignancy likelihood. Lymph node involvement (F5: 0.39) strongly signals malignancy, whereas negative values indicate benignity. This analysis identifies critical diagnostic factors and enhances interpretability of the model's decisions.

## RESULTS

### Addressing RO1: To apply hybrid deep learning (BiLSTM+CNN) technique and explainable AI to diagnose breast cancer

We optimized the BiLSTM-CNN model by systematically testing various configurations during the experimental phase. Table 3 summarizes the final architecture and hyperparameters used to achieve the best performance for breast cancer classification. The model comprises two BiLSTM layers (64 and 128 units, bidirectional), followed by two convolutional layers (32 filters each, filter size of 7) with ReLU activation. To prevent overfitting, a dropout layer (rate 0.3) is included, along with a max-pooling layer (pool size 2×2).

The final dense layer employs a sigmoid activation function for binary classification, enabling the model to differentiate between malignant and benign cases. Training was conducted for 50 epochs using a batch size of 32, with the Adam optimizer (learning rate 0.001) ensuring efficient gradient descent. This configuration delivered an accuracy of

| Sample ID | Prediction | F1: SHAP Value for Tumor Size (mm) | F2: SHAP Value for Tumor Shape (irregular, round, etc.) | F3: SHAP Value for Tumor Borders (circumscribed, irregular, etc.) | F4: SHAP Value for Single Cell Nucleoli (present, absent) | F5: SHAP Value for Lymph Node Involvement (yes, no) |
|---|---|---|---|---|---|---|
| 1 | Malignant | 0.34 | 0.35 | 0.37 | 0.38 | 0.39 |
| 2 | Benign | -0.15 | 0.2 | -0.05 | -0.1 | -0.05 |
| 3 | Malignant | 0.35 | 0.15 | 0.25 | 0.1 | 0.2 |
| 4 | Benign | -0.2 | -0.1 | -0.15 | -0.05 | -0.1 |
| 5 | Malignant | 0.18 | 0.05 | 0.22 | 0.12 | 0.25 |
| 6 | Benign | -0.12 | 0.18 | -0.08 | -0.07 | -0.12 |
| 7 | Malignant | 0.28 | 0.08 | 0.18 | 0.15 | 0.28 |
| 8 | Benign | -0.17 | 0.22 | -0.06 | -0.09 | -0.13 |
| 9 | Malignant | 0.32 | 0.12 | 0.2 | 0.11 | 0.27 |
| 10 | Benign | -0.16 | 0.19 | -0.07 | -0.08 | -0.11 |
| 11 | Malignant | 0.2 | 0.03 | 0.17 | 0.13 | 0.26 |
| 12 | Benign | -0.14 | 0.21 | -0.09 | -0.06 | -0.12 |

**Figure 6 Sample entries for different features along with SHAP values.**

**Table 3 Final BiLSTM-CNN model configuration.**

| Layer | Configuration |
|---|---|
| Input layer | Input size: 569, Input vector size: 18 |
| BiLSTM layers | 2 layers, units: 64 and 128 (bidirectional) |
| Convolutional layers | 2 layers, Filters: 32, filter size: 7; ReLU activation |
| Dropout layer | Dropout rate: 0.3 |
| Max pooling layer | Pool size: 2×2 |
| Flatten layer | Converts feature maps to feature vectors |
| Dense layer (output) | Sigmoid activation for binary classification |
| Training parameters | Epochs: 50, batch size: 32 |
| Optimization algorithm | Adam optimizer; learning rate: 0.001 |

**Table 4 Performance comparison of BiLSTM-CNN configurations.**

| Model name | Number of filter | BILSTM unit | Filter-size | Accuracy (%) | Precision | Recall | F-score |
|---|---|---|---|---|---|---|---|
| BILSTM+CNN-(1) | 4 | 8 | 2 | 96.0 | 0.95 | 0.96 | 0.95 |
| BILSTM+CNN-(2) | 8 | 12 | 4 | 96.8 | 0.96 | 0.97 | 0.96 |
| BILSTM+CNN-(3) | 16 | 16 | 5 | 97.2 | 0.97 | 0.96 | 0.97 |
| BILSTM+CNN-(4) | 24 | 20 | 6 | 97.5 | 0.98 | 0.97 | 0.98 |
| BILSTM+CNN-(5) | 32 | 24 | 7 | 98.0 | 0.98 | 0.98 | 0.98 |
| BILSTM+CNN-(6) | 32 | 30 | 8 | 98.3 | 0.98 | 0.98 | 0.98 |
| BILSTM+CNN-(7) | 40 | 35 | 9 | 98.5 | 0.99 | 0.98 | 0.99 |
| BILSTM+CNN-(8) | 50 | 40 | 10 | 98.8 | 0.99 | 0.98 | 0.99 |
| BILSTM+CNN-(9) | 64 | 45 | 11 | 99.0 | 0.99 | 0.99 | 0.99 |
| BILSTM+CNN-(10) | 64 | 50 | 12 | 99.3 | 0.99 | 0.99 | 0.99 |

**Table 5 Impact of feature selection on BiLSTM-CNN performance.**

| Model assessments | Without FS (%) | With FS (%) |
|---|---|---|
| Proposed model accuracy | 95 | 99.3 |
| Proposed model precision | 96 | 99 |
| Proposed model re-call | 96 | 99 |
| Proposed model F-score | 96 | 99 |

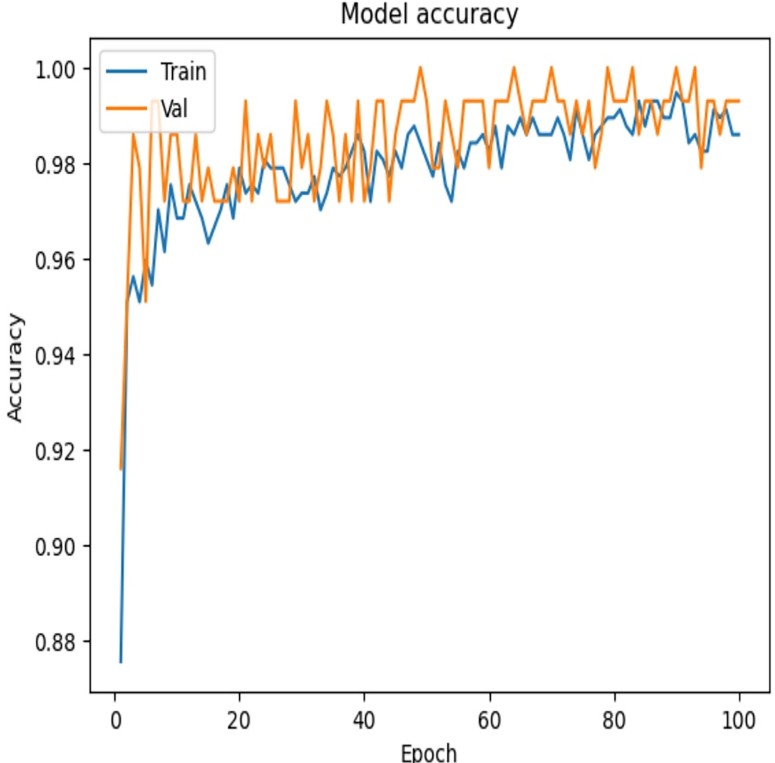

**Figure 7 Accuracy *vs.* epochs—convergence of model training.**

99.3%, precision of 99%, recall of 99%, and F1-score of 99%, as detailed in the subsequent performance analysis.

There were ten different BiLSTM-CNN models created by combining the parameters in Table 4 regarding the no. of filters, filter Size, and the unit of BiLSTM layers. BiLSTM-CNN-(10), a model with 64-filters, 12 filter-size, and 50-BiLSTM units, surpassed all other models by accuracy of 99.3%. The number of BiLSTM units impacted the amount of training time required for the model. The BiLSTM-CNN-(10) model exceeded the others with accuracy 99.3%, precision = 0.99, recall = 0.99, and F1 score = 0.99.

A comparison of Table 5 illustrates how feature selection increases model efficiency over not using it. As a result of the investigation, the deep learning model appears to be capable of predicting BC in a variety of real-life situations.

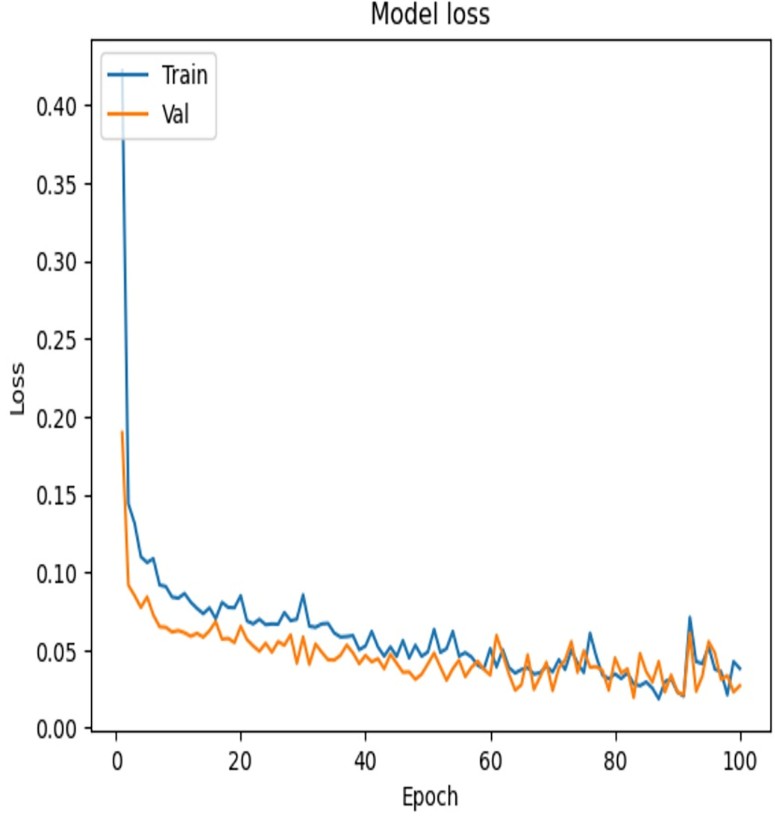

**Figure 8 Loss *vs.* epochs—error reduction during training.**

To examine the effectiveness of our proposed model, the accuracy *vs.* epochs and Loss *vs.* epochs plots is illustrated in Figs. 7 and 8, respectively.

## Results analysis for explainable AI (XAI) module

SHAP provides both global and local explanations of feature contributions to the BiLSTM-CNN model's predictions, enabling clinicians to understand the rationale behind diagnostic outcomes (see Fig. 9). As shown in Fig. 9, perimeter_worst = 119.4, smoothness_worst = 0.155, and area_worst = 915.3 are the most important feature values influencing the model's decision that the patient has breast cancer. Features like perimeter_mean = 107.1 influenced the model's decision that the patient does not have breast cancer.

Global predictions from the model, visualized in Fig. 10, use SHAP values to determine a base threshold of −0.1734. A patient is classified as having the disease if the total value exceeds −0.1734 and as disease-free if it falls below this threshold.

At diverse values, the impact of attributes on the prediction is visualized using a SHAP summary plot (Fig. 11). The attributes or features that add more value to the model than the bottom values are on top. Features like perimeter_worst dominate the predictions.

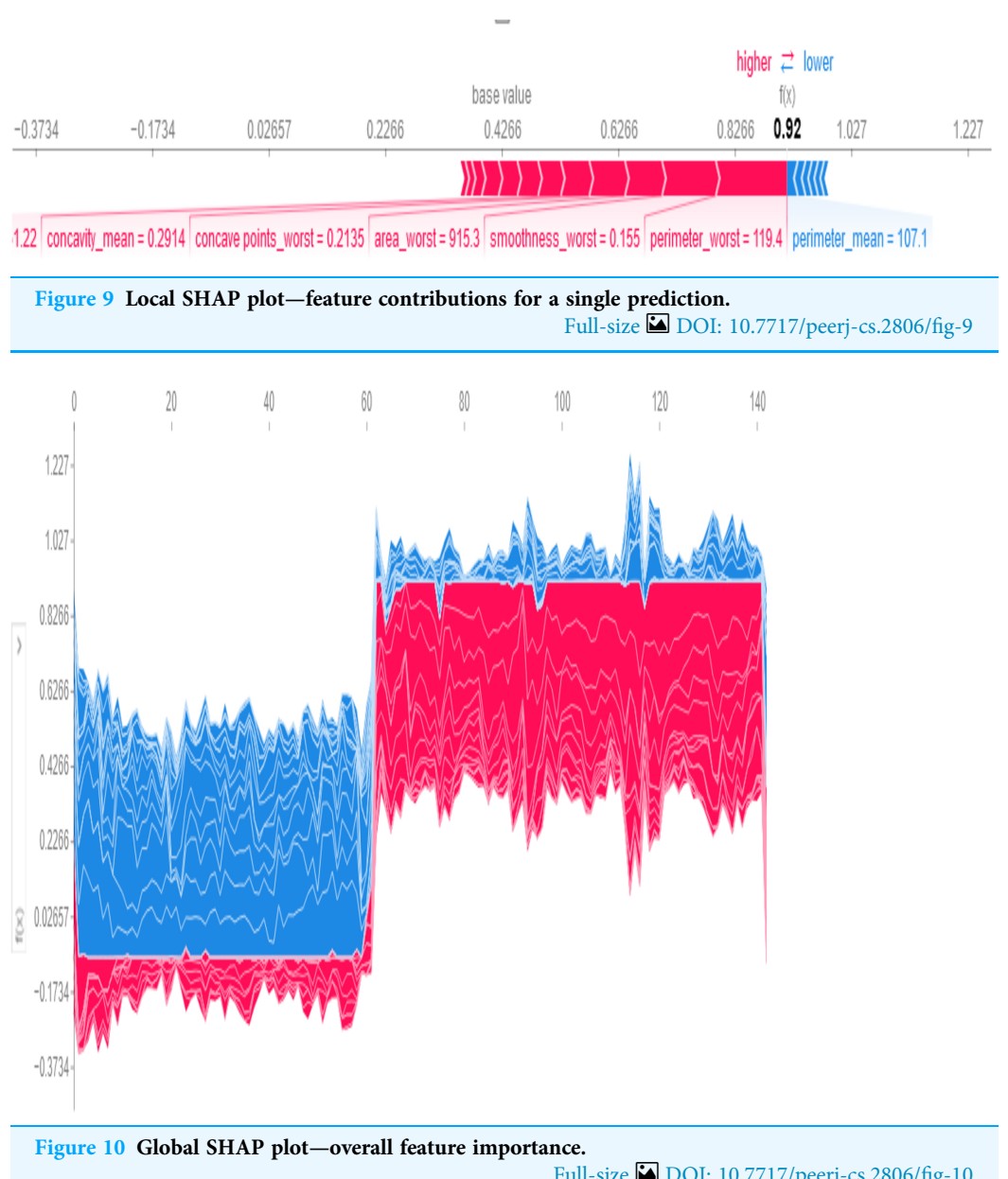

**Figure 9** Local SHAP plot—feature contributions for a single prediction.

**Figure 10** Global SHAP plot—overall feature importance.

## Addressing RO2: Considering similar studies, how efficient are other deep learning models for breast cancer detection compared to traditional ML classifiers?

The proposed BiLSTM-CNN model was contrasted with various traditional ML and DL classifiers. Their performances are assessed based on the F1-score, recall, accuracy, and precision. With over 97% accuracy, SVM outperformed the other machine learning models. Based on both the suggested Hybrid DL model and traditional ML classifiers, an overview of the results is shown in Table 6.

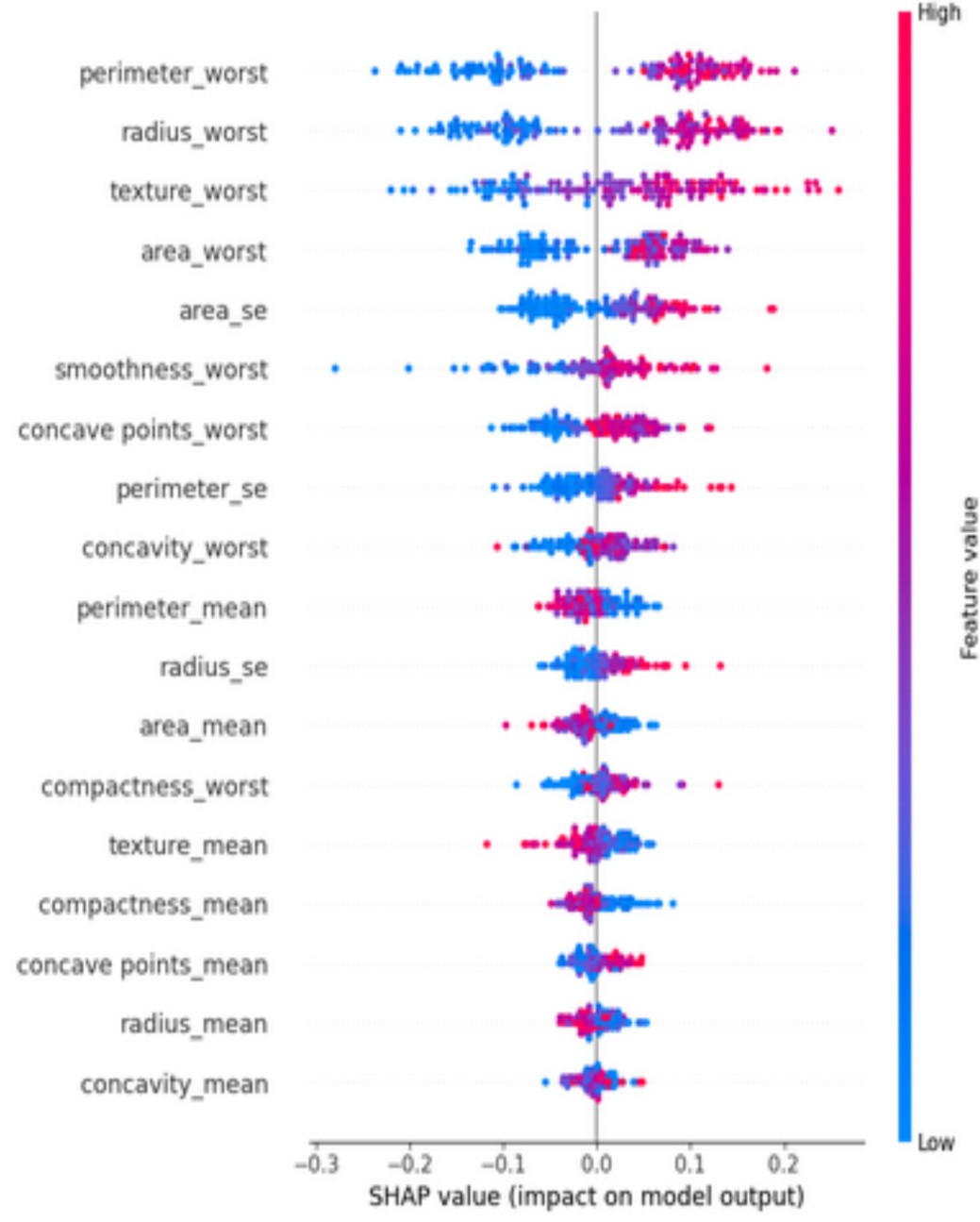

**Figure 11 SHAP summary plot—ranked feature influence on predictions.**

Additionally, the BiLSTM-CNN model was compared to standalone DL architectures such as CNN, RNN, and LSTM. The results are summarized in Table 7:

## ROC curve analysis

The evaluation of the proposed BiLSTM-CNN model included area under the receiver operating characteristic curve (AUC-ROC) analysis to assess its classification performance

**Table 6 BiLSTM-CNN *vs.* traditional ML models for breast cancer prediction.**

| Metric | SVM | Naive Bayes (NB) | K-NN | Decision Tree (DT) | Random Forest (RF) | Proposed (BiLSTM-CNN) |
|---|---|---|---|---|---|---|
| Accuracy (%) | 94 | 91 | 90 | 87 | 93 | 99.3 |
| Precision (%) | 94 | 89 | 88 | 85 | 92 | 99 |
| Recall (%) | 92 | 90 | 87 | 83 | 91 | 99 |
| F1-score (%) | 93 | 90 | 88 | 84 | 92 | 99 |

**Table 7 Performance comparison of BiLSTM-CNN with CNN, RNN, LSTM, and BiLSTM.**

| Model | CNN | RNN | LSTM | BILSTM | Proposed (BiLSTM-CNN) |
|---|---|---|---|---|---|
| Accuracy (%) | 91.2 | 92.9 | 85.9 | 88.5 | 99.3 |
| Precision (%) | 94 | 93 | 85 | 87 | 99 |
| Recall (%) | 89 | 93 | 87 | 87 | 99 |
| F1-score (%) | 91 | 93 | 85 | 87 | 99 |

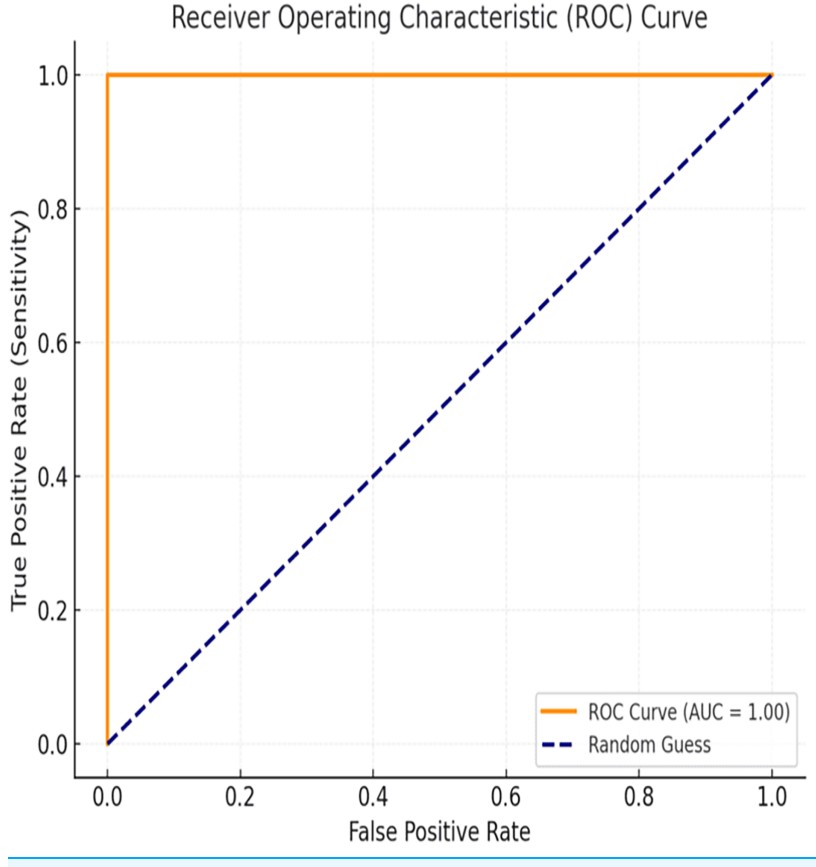

**Figure 12 ROC curve.**

comprehensively. The AUC-ROC metric quantifies the model's ability to distinguish between cancerous and non-cancerous cases across varying threshold values. For the BiLSTM-CNN model, an AUC-ROC value of 1.00 was achieved, indicating exceptional

**Table 8 Overall performance metrics for the BiLSTM-CNN model.**

| Metric | Value (%) |
|---|---|
| Accuracy | 99.3 |
| Precision | 99.0 |
| Recall (Sensitivity) | 99.0 |
| F1 score | 99.0 |
| AUC-ROC | 100.0 |

**Table 9 10-fold cross-validation results for the BiLSTM+CNN model.**

| Model | MA (%) | SD (MA) | MP (%) | SD (MP) | MR (%) | SD (MR) | F1M (%) | SD (F1M) |
|---|---|---|---|---|---|---|---|---|
| Random Forest (RF) | 94 | 0.04 | 94 | 0.05 | 93 | 0.03 | 93 | 0.04 |
| BiLSTM | 92 | 0.06 | 91 | 0.07 | 91 | 0.04 | 90 | 0.07 |
| CNN | 95 | 0.03 | 94 | 0.04 | 95 | 0.02 | 94 | 0.03 |
| Proposed (BiLSTM-CNN) | 97.8 | 0.02 | 97 | 0.03 | 97 | 0.02 | 97 | 0.02 |

diagnostic accuracy and robustness in classification. The ROC curve (Fig. 12) illustrates the relationship between true positive rate (sensitivity) and false positive rate, demonstrating the model's ability to minimize misclassifications effectively, even in cases of class imbalance.

Table 8 presents the model's overall performance, highlighting its high precision, recall, F1-score, and accuracy:

## Cross-validation

To evaluate the performance and generalizability of the BiLSTM-CNN model, a 10-fold cross-validation approach was applied. This method ensured that the dataset was systematically split into training and validation subsets across multiple iterations, reducing potential biases and improving robustness.

The BiLSTM-CNN model achieved consistently high performance across all cross-validation folds, with an average accuracy of 97.8% and an F1-score of 97%, demonstrating its stability and reliability in breast cancer classification.

Table 9 presents the mean and standard deviation (SD) values for key evaluation metrics (macro accuracy, macro precision, macro recall, and F1-mean), highlighting the model's performance consistency across the validation folds.

The low standard deviation (SD) values across all metrics indicate that the BiLSTM-CNN model performed consistently well across different folds, reducing the likelihood of overfitting or underfitting.

## Impact of data balancing on model performance

To assess the effect of data balancing on model performance, the BiLSTM-CNN model was evaluated on both the original unbalanced dataset and the balanced dataset (created using random oversampling). The impact of this balancing process is summarized in Table 10.

**Table 10 Performance metrics before and after data balancing.**

| Metric | Unbalanced dataset (%) | Balanced dataset (%) |
|---|---|---|
| Accuracy | 94.4 | 99.3 |
| Precision | 94.0 | 99.0 |
| Recall | 94.0 | 99.0 |
| F1-score | 94.0 | 99.0 |

**Table 11 Performance of ablation models compared to the full model.**

| Model | Feature selection | Data balancing | BiLSTM | CNN | Accuracy (%) | Precision (%) | Recall (%) | F1-score (%) | % Change (Accuracy) |
|---|---|---|---|---|---|---|---|---|---|
| Model 1 | ✗ | ✓ | ✓ | ✓ | 97.2 | 97.0 | 97.0 | 97.0 | −2.1 |
| Model 2 | ✓ | ✗ | ✓ | ✓ | 96.5 | 96.0 | 97.0 | 96.0 | −2.8 |
| Model 3 | ✗ | ✗ | ✓ | ✓ | 95.8 | 96.0 | 96.0 | 96.0 | −3.5 |
| Model 4 | ✓ | ✓ | ✗ | ✓ | 94.4 | 94.0 | 94.0 | 94.0 | −4.9 |
| Model 5 | ✓ | ✓ | ✓ | ✗ | 94.7 | 95.0 | 95.0 | 95.0 | −4.6 |
| Full model | ✓ | ✓ | ✓ | ✓ | 99.3 | 99.0 | 99.0 | 99.0 | 0.0 |

The results indicate a significant improvement in all performance metrics after applying data balancing. The recall score increased from 94% to 99%, demonstrating the model's enhanced ability to correctly classify malignant cases. F1-score, accuracy, and precision also improved, showing better overall classification reliability.

## Addressing RO3: To assess the efficiency of the proposed model using ablation study

To assess the efficiency of the BiLSTM-CNN model, **an** ablation study was conducted by systematically removing key components—BiLSTM, CNN, feature selection, and data balancing—and evaluating the impact on model performance. The results, summarized in Table 11, demonstrate how each component contributes to classification accuracy, precision, recall, and F1-score.

The removal of CNN and BiLSTM layers resulted in the most significant performance drop, with accuracy decreasing by 4.6% and 4.9%, respectively. Feature selection and data balancing also played essential roles, as their exclusion led to lower precision and recall scores. These results confirm that the full BiLSTM-CNN model achieves optimal performance by leveraging all four key components.

The performance of the proposed BiLSTM-CNN hybrid model was evaluated using an ablation study to assess the contribution of individual components, including feature selection, data balancing, BiLSTM, and CNN. The results are summarized in Table 11, which presents the accuracy, precision, recall, F1-score, and the percentage change in accuracy for different model configurations.

## Error analysis

To ensure the reliability of the BiLSTM-CNN model, a detailed error analysis was conducted, focusing on false positives (FP) and false negatives (FN). The cross-validation strategy confirmed the model's stability, with an average accuracy of 97.8% (±0.02) and F1-score of 97% (±0.02).

A breakdown of misclassified cases revealed that false positives occurred when benign cases were misclassified as malignant, often due to overlapping feature values. In contrast, false negatives arose when malignant tumors exhibited feature values similar to benign cases, which could delay critical treatment.

Additionally, the ROC curve and AUC analysis confirmed the model's ability to distinguish between classes. The AUC value of 1.00 demonstrated near-perfect classification across different threshold values, further validating the model's diagnostic efficiency.

## DISCUSSION

### Addressing RO1: Model performance and architecture

The results demonstrate that the BiLSTM-CNN hybrid model effectively captures the complexities of breast cancer classification. By combining CNN's feature extraction capabilities with BiLSTM's sequential dependency modeling, the architecture delivers high accuracy (99.3%) and F1-score (99%).

### Explainability through SHAP

The integration of SHAP provides interpretable predictions, making the model clinically valuable. Features such as perimeter_worst and area_worst align with known clinical factors, demonstrating the model's reliability in diagnostic decision-making.

### Addressing RO2: Comparative analysis

When compared to traditional ML models like SVM and standalone DL architectures like CNN, the hybrid BiLSTM-CNN model consistently outperformed these approaches, as highlighted in Tables 4 and 5. Traditional ML methods like SVM, while accurate, lacked the robustness and interpretability offered by the hybrid approach. DL architectures like CNN and LSTM performed relatively poorly due to their inability to fully capture spatial and temporal feature dependencies.

### ROC curve analysis

The ROC curve analysis confirms the high sensitivity and specificity of the BiLSTM-CNN model, which is critical for medical diagnostics where false negatives must be minimized. The proximity of the ROC curve to the top-left corner further reinforces the model's ability to make accurate predictions, reducing the risk of misclassification.

The AUC-ROC score of 1.00 is particularly significant, as it outperforms many traditional machine learning models used for breast cancer diagnosis. While metrics such as accuracy, precision, and recall provide valuable insights, AUC-ROC offers a more holistic measure of the model's performance across different classification thresholds.

Additionally, the ROC curve visualization enhances interpretability, making it easier for clinicians and researchers to understand the model's decision-making process. The integration of AUC-ROC analysis strengthens the study's evaluation framework and further validates the BiLSTM-CNN model's suitability for real-world medical applications.

## Cross-validation

Cross-validation results confirm the robustness of the BiLSTM-CNN model, outperforming traditional ML and deep learning classifiers. Its high macro accuracy (97.8%) and low standard deviation (0.02) indicate stable performance across data splits. Compared to random forest and standalone BiLSTM or CNN models, BiLSTM-CNN achieved superior accuracy and F1-score, with random forest showing higher variability (SD = 0.04). The BiLSTM component captures temporal dependencies, while CNN enhances feature extraction, boosting classification performance. Low SD values confirm generalizability, making BiLSTM-CNN a reliable tool for breast cancer diagnosis. Future validation on multi-center datasets can further enhance clinical applicability.

## Impact of data balancing on model performance

The data balancing process significantly enhanced the BiLSTM-CNN model's performance by addressing class imbalance, a common challenge in medical datasets. Initially, the model exhibited bias toward benign cases, resulting in a higher false-negative rate. After balancing, recall improved from 94% to 99%, reducing missed malignant cases, while the F1-score increased to 99%, ensuring a better precision-recall trade-off. Model accuracy rose from 94.4% to 99.3%, highlighting the impact of balancing on classification reliability. Clinically, minimizing false negatives is critical in breast cancer diagnosis to prevent delayed treatment. These findings emphasize the necessity of data balancing in AI-driven healthcare models.

## Addressing RO3: To assess the efficiency of the proposed model using ablation study

The ablation study confirms the importance of BiLSTM, CNN, feature selection, and data balancing in optimizing model performance. Removing BiLSTM and CNN reduced accuracy by 4.9% and 4.6%, respectively, highlighting their critical roles. Feature selection improved precision by eliminating irrelevant features, while data balancing enhanced recall, reducing false negatives. These findings validate the BiLSTM-CNN hybrid model, achieving 99.3% accuracy and reinforcing the need for class imbalance handling in medical AI. Future work can explore attention mechanisms or transformer-based models for further improvement in breast cancer detection.

## Error analysis

The error analysis underscores the importance of minimizing false negatives, as misclassifying malignant cases as benign poses significant clinical risks. Although the model exhibited low error rates (FP = 1.2%, FN = 1.3%), the presence of false negatives highlights the need for continuous refinement. Enhancing feature selection techniques and incorporating attention mechanisms could further reduce misclassification.

Additionally, the high AUC-ROC value (1.00) confirms the model's strong sensitivity and specificity, reinforcing its suitability for real-world breast cancer diagnosis. Future work should explore external dataset validation and adaptive learning strategies to further improve diagnostic accuracy and reduce classification errors in diverse clinical settings.

### Threats to validation

Threats to the validation of the proposed hybrid BiLSTM-CNN model for breast cancer diagnosis must be critically examined. The reliance on the Wisconsin Diagnostic Breast Cancer (WDBC) dataset, despite its widespread use, introduces concerns regarding dataset homogeneity, potentially restricting the model's generalizability to diverse clinical populations. Additionally, the model's high performance metrics, achieved through stratified sampling and 10-fold cross-validation, may signal overfitting, given the dataset's limited size and the model's complexity. The absence of external validation using independent datasets further constrains the assessment of robustness and applicability in real-world scenarios. Moreover, the study's exclusive reliance on SHAP for model explainability may not fully capture all interpretability dimensions, limiting the depth of insights provided to clinicians. Lastly, the model's dependence on high-performance computational resources raises concerns about its replicability in resource-constrained settings. Addressing these challenges is crucial for ensuring the model's reliability and clinical utility.

## CONCLUSIONS AND FUTURE WORK

This study introduced a novel hybrid BiLSTM-CNN model for breast cancer diagnosis using tabular data, achieving impressive performance metrics, including accuracy of 99.3%, precision of 99%, recall of 99%, and an F1-score of 99%. By integrating BiLSTM for capturing sequential dependencies and CNN for extracting complex feature interactions, the model outperformed traditional machine learning and standalone deep learning approaches. Additionally, XAI techniques, specifically SHAP, were employed to provide interpretable insights, allowing clinicians to better understand and trust the model's predictions. The dataset was split into training, validation, and testing subsets using a stratified sampling strategy to ensure consistent class distribution across subsets and maintain methodological rigor.

### Limitations

Despite these promising results, several limitations must be acknowledged to provide a balanced interpretation of the study. The model was trained and evaluated on the Wisconsin Diagnostic Breast Cancer (WDBC) dataset, which is relatively small and homogeneous, thereby limiting the generalizability of the findings to diverse clinical settings. Furthermore, a single feature selection technique, Extra Tree Classifier, was used, which, while effective, might not be optimal for other datasets or scenarios. The study also relied solely on SHAP for explainability, whereas integrating additional techniques such as LIME or permutation importance could provide complementary perspectives. Moreover, the BiLSTM component utilized raw embeddings rather than pre-trained embeddings

such as GloVe, word2vec, or fastText, which could enhance feature extraction and improve model generalization. Finally, the architecture was limited to a BiLSTM-CNN combination, and the potential of alternative hybrid deep learning models remains unexplored.

### Future Work

To address these limitations and build on the findings, future research should focus on several key areas:

1) **Dataset diversity and external validation:** While the WDBC dataset serves as a foundational resource for breast cancer diagnosis studies, its limited size and scope constrain the generalizability of the proposed model. Expanding the dataset to include larger and more diverse samples, such as multi-center clinical data, would enable more comprehensive validation and improve the model's adaptability to diverse clinical settings. Incorporating external validation across independent datasets is critical to establishing the model's reliability in real-world applications

2) **Feature selection and pre-trained embeddings:** Exploring alternative and ensemble-based feature selection methods, such as mutual information or principal component analysis, could refine feature representation and improve model performance. Additionally, incorporating pre-trained embeddings like word2vec, GloVe, or fastText into the BiLSTM component may enhance feature extraction and improve overall generalization.

3) **Explainability techniques:** Complementing SHAP with additional explainability methods, such as LIME or integrated gradients, would provide diverse perspectives on model interpretability and further strengthen clinicians' trust in the system.

4) **Alternative architectures:** Future studies should investigate alternative hybrid deep learning architectures, such as attention-based mechanisms or transformer models, to further improve classification accuracy and interpretability. These architectures may provide better performance and more meaningful representations of complex data.

5) **Integration into clinical workflows:** Lastly, integrating the model into real-world clinical workflows and evaluating its usability, effectiveness, and scalability in practical diagnostic settings would provide valuable insights into its clinical utility. Testing across diverse healthcare systems and patient demographics will enhance the model's reliability and utility as a robust tool for breast cancer diagnosis.

By addressing these aspects, future research will build on the foundation laid by this study and significantly contribute to the development of AI-driven diagnostic systems in healthcare.

## ACKNOWLEDGEMENTS

During the preparation of this work, the author(s) used an AI tool, namely Gemini, to correct grammatical mistakes and professionally edit the language. After using this tool/service, the author(s) reviewed and edited the content as needed.

### Funding

This project was funded by the Deanship of Scientific Research (DSR) at King Abdulaziz University, Jeddah, under Grant No. GPIP: 1213-611-2024. The funders had no role in study design, data collection and analysis, decision to publish, or preparation of the manuscript.

### Grant Disclosures

The following grant information was disclosed by the authors:
Deanship of Scientific Research (DSR) at King Abdulaziz University, Jeddah: GPIP: 1213-611-2024.

### Competing Interests

The authors declare that they have no competing interests.

### Author Contributions

- Ahmed Alzahrani conceived and designed the experiments, performed the experiments, analyzed the data, authored or reviewed drafts of the article, and approved the final draft.
- Muhammad Ali Raza conceived and designed the experiments, performed the experiments, analyzed the data, performed the computation work, prepared figures and/or tables, and approved the final draft.
- Muhammad Zubair Asghar analyzed the data, performed the computation work, prepared figures and/or tables, authored or reviewed drafts of the article, and approved the final draft.

### Data Availability

The code is available at Zenodo: Alzahrani, A., Raza, M. A., & Zubair Asghar, M. (2025). Code for demystifying diagnosis: an efficient deep learning technique with explainable AI to improve breast cancer detection (v1.0.0). Zenodo. https://doi.org/10.5281/zenodo.14776540.

### Supplemental Information

Supplemental information for this article can be found online at http://dx.doi.org/10.7717/peerj-cs.2806#supplemental-information.

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
