# Peer review of "Demystifying diagnosis: an efficient deep learning technique with explainable AI to improve breast cancer detection"

_PeerJ Computer Science, doi:10.7717/peerj-cs.2806_

## Round 0.1 · original submission · Major Revisions

The paper can be improved accordingly to reviewers suggestions.

Reviewer 1 ·

Basic reporting

The article is written in mostly professional English, though improvements in clarity and readability are recommended. Some sentences are overly complex, which may impede comprehension. A thorough proofreading and potentially professional editing would enhance the clarity and flow of the text.
Background and Literature Context: While the background on breast cancer diagnosis is relevant, the literature review would benefit from including more recent studies on explainable AI (XAI) in healthcare. This would better position the work within the context of current advancements and clarify its unique contribution.
Structure and Visuals: The overall structure aligns with academic standards, though figures, particularly SHAP visualizations, should be provided in higher resolution to improve interpretability. Captions could be more detailed to help readers understand the significance of each visual without needing to refer back to the main text.
Reproducibility and Data Sharing: Although the authors use a publicly available dataset, reproducibility could be enhanced by sharing their code and implementation details on a platform like GitHub. This would allow other researchers to replicate and validate the model's results.
Definitions and Formal Results: Some technical terms, such as SHAP values and BiLSTM, are not clearly defined on first mention. Providing definitions and ensuring all terms are accessible would improve the paper’s self-contained nature. Additionally, more formal derivations or proofs, especially for the results from SHAP and performance comparisons, would add rigor to the findings.

Experimental design

The study demonstrates a relevant research question, aiming to enhance breast cancer diagnosis through a hybrid BiLSTM-CNN model with SHAP-based explainability, which aligns with the journal’s aims and scope. The research question addresses a meaningful gap, highlighting the need for interpretable AI in medical diagnosis. However, several areas require improvement to meet rigorous standards:
1. Detail in Methodology: While the general framework of the BiLSTM-CNN model and SHAP analysis is presented, there are critical gaps in methodological details, which may hinder reproducibility. The authors should provide comprehensive details, particularly in model hyperparameters, training configurations, and the specific implementation of SHAP, to ensure other researchers can replicate the study.
2. Choice of CNN in Non-Image Data: The use of CNNs in a tabular dataset lacks sufficient justification, as CNNs are traditionally applied in spatial data contexts. The authors should clarify why CNN layers were chosen over more suitable architectures for tabular data, such as fully connected layers or other feature extraction methods, and discuss any modifications made to adapt CNNs for this purpose.
3. Dataset Limitations: The study relies solely on the Wisconsin Breast Cancer Diagnostic dataset, which, while a well-known source, is relatively small and may limit generalizability. Using additional datasets or including cross-validation strategies to account for dataset size limitations would strengthen the results. At minimum, the authors should discuss the impact of this limitation on model performance and applicability.
4. Explainability Techniques: While SHAP values are used for interpretability, the rationale for choosing SHAP over other explainability techniques (such as LIME or permutation importance) is not discussed. Given the importance of interpretability in this study, a comparative rationale would clarify the methodological choice and improve the study’s rigor.
5. Code Availability: To promote reproducibility, the authors are strongly encouraged to share their code and implementation details in a public repository. This would uphold rigorous standards and support other researchers in verifying and building upon this work.

Validity of the findings

The study offers an innovative approach by integrating BiLSTM-CNN and SHAP explainability for breast cancer diagnosis. However, to fully meet publication standards, several issues regarding the validity and robustness of the findings should be addressed:
1. Impact and Novelty: Although the authors present a hybrid model that is promising in the field of breast cancer diagnosis, the specific novelty of using BiLSTM-CNN in a non-image context is not thoroughly justified. More emphasis should be placed on the unique contributions of this hybrid approach compared to previous studies, particularly regarding why this specific architecture improves interpretability and diagnostic accuracy in tabular datasets. Clarifying this would better support the study's impact.
2. Data Robustness and Statistical Validity: The findings are derived from a single, relatively small dataset (Wisconsin Breast Cancer Diagnostic), which may limit generalizability. There is also a reliance on a high accuracy score without presenting robust statistical validation methods. It would improve the study’s rigor to include a more detailed statistical analysis, such as confidence intervals, significance testing, or error analysis, to confirm that the results are not overly optimistic or dataset-specific. Additionally, cross-validation could better ensure the stability of the reported metrics.
3. Control for Data Imbalance: The study mentions addressing class imbalance through oversampling but lacks details on how this was validated to avoid potential bias or overfitting. The authors should consider reporting performance metrics beyond accuracy, such as AUC-ROC, precision-recall curves, and F1 scores, especially since medical datasets often benefit from such additional evaluation metrics due to potential class imbalances.
4. Reproducibility and Transparency: To substantiate the study's findings, it is essential that the authors share their implementation code and data preprocessing steps publicly. This would support meaningful replication and validate that the reported improvements are not due to data-specific tuning.
5. Clarity of Conclusions: The conclusions are generally aligned with the research question; however, they should explicitly discuss limitations, particularly regarding dataset size and generalizability. It is also recommended that the authors temper any claims about the model's general utility in clinical settings, given the limited dataset used and the specific context of the Wisconsin dataset. A more balanced discussion would improve the clarity and reliability of the conclusions.

Additional comments

Comments to the Author
The manuscript, titled "Demystifying Diagnosis: An Efficient Deep Learning Technique with Explainable AI to Improve Breast Cancer Detection," presents a promising approach by integrating BiLSTM and CNN, coupled with SHAP for model interpretability in breast cancer detection. This research is timely and addresses a critical need for interpretability alongside predictive accuracy in medical diagnostics. The following comments are provided for further improvement:
1. Basic Reporting
o Language and Terminology: While generally well-written, a careful review of grammar and terminology (e.g., defining "Explainable AI" and "SHAP" on first use) would enhance readability for diverse audiences.
o Redundant Phrasing: Certain sections repeatedly mention motivations for using XAI. Condensing these to focus directly on the contributions of this hybrid model could improve clarity.
o Figures and Tables: Adding more context to figure captions, especially within Table 4, would clarify the significance of the comparisons made.
2. Experimental Design
o Dataset and Data Splitting: The Wisconsin dataset is appropriate but limited in diversity. Clarifying data splitting strategies (e.g., whether stratification was used) would enhance the methodological rigor. Expanding future studies to additional datasets or external validation would be beneficial.
o Feature Selection: Including a rationale for choosing the Extra Trees Classifier over alternatives (e.g., Lasso, PCA) would strengthen the methodology. Any statistical validation for the chosen features, such as correlation metrics, would further support this choice.
o Data Balancing: Detailing metrics before and after balancing would illustrate its impact on model performance.
3. Methodology and Model Architecture
o BiLSTM-CNN Hybrid Model: Describing layer configurations, dropout rates, and other hyperparameters in a summarized table would improve reproducibility.
o Evaluation Metrics: While accuracy, precision, recall, and F1 scores are presented, additional metrics like AUC-ROC would add depth, particularly given the diagnostic nature of the model.
o Interpretability: Expanding on SHAP’s limitations and possibly comparing it briefly with other interpretability methods, such as LIME, would strengthen the rationale for this choice.
4. Results and Analysis
o Model Comparison: While thorough, the model comparisons could benefit from discussing how BiLSTM-CNN leverages temporal and spatial features for improved performance over traditional models.
o Cross-Validation: Reporting the standard deviation across folds would clarify model robustness.
o Ablation Study: Adding insights into how each component (BiLSTM, CNN, Feature Selection, Data Balancing) influences different metrics could enrich the analysis.
5. Conclusions and Limitations
o Limitations: Explicitly discussing limitations, such as dataset constraints, choice of feature selection and XAI method, would enhance transparency regarding generalizability.
o Future Directions: Mentioning the potential use of pre-trained embeddings (e.g., GloVe, word2vec) could further improve model performance.
Additional Recommendations
• Public Code Availability: Sharing the code on a public platform (e.g., GitHub) with instructions would support reproducibility.
• Glossary: Briefly defining technical terms or including a glossary could make the paper more accessible.

Reviewer 2 ·

Basic reporting

Dear authors,

Please make the changes and resubmit the manuscript.

1. Mention the contributions of your research in points in the introduction section. Also add a graphical abstract regarding breast cancer.

Literature review about XAI is not sufficient. Please include latest papers from 2024.

Results and Discussion section should be seperate. I admire the results and ablation study!

Discussion section should be improved. You could look into the above papers to have a strong discussion section

Add threat to validation section along with limitations and future research directions.

Good luck!

Experimental design

Okay. Minor changes.

Validity of the findings

Okay. Minor changes.

Additional comments

Please refer the above comments.

Reviewer 4 ·

Basic reporting

Suggestions:
1. Refine the use of technical jargon and rephrase complex sentences to improve readability and ensure clarity.
2. Expand the literature review to include more recent studies to showcase the relevance and novelty of your approach in the context of the latest developments in the field.

Experimental design

Comments:

1. The experimental design is sound, with clear definitions of the research question and objectives. The methods section is detailed, providing enough information for reproducibility.

2. The dataset used, while appropriate, is limited in scope, and the implications of this limitation on the generalizability of the findings are not adequately discussed.

Suggestion:
• Integrate more comprehensive statistical analysis to support the conclusions drawn, such as p-values, confidence intervals, and discussion on data balance.

Validity of the findings

No Comment

Additional comments

The manuscript presents valuable research with potential impact in the field of breast cancer detection using explainable AI.

However, to meet the publication standards fully, it is essential to enhance the clarity and accessibility of the language, update and deepen the literature review, broaden the dataset applicability, and solidify the statistical underpinnings of the study.

Addressing these points will significantly strengthen the paper and ensure that it makes a substantial contribution to the scientific community.

Reviewer 5 ·

Basic reporting

Strengths:

The figures and tables are well-labeled, contributing to the manuscript's clarity.

Weaknesses:

The introduction lacks depth in references. It would benefit from citing additional recent and relevant studies to provide a stronger foundation for the research problem.
The hybrid BiLSTM+CNN technique is introduced in Section B without sufficient context or explanation of why this specific architecture is suitable for the dataset and problem at hand. A clear problem formulation and justification are missing.

Suggestions for improvement:

Enhance the introduction by integrating more references to recent studies that align with the manuscript’s scope, focusing on XAI, hybrid deep learning models, and breast cancer detection.
Clearly state the rationale behind using the BiLSTM+CNN model and how it compares to other techniques in the introduction or problem formulation sections.

Experimental design

Strengths:

-The use of SHAP for explainability is a commendable addition that aligns with the goals of XAI.

Weaknesses:

-The methods are not described in detail, which does not allow for reproducibility to a certain extent.

- Feature Selection and Cross-Validation: the feature selection procedure appears to be conducted outside of the cross-validation folds, potentially introducing data leakage (double-dipping bias). This undermines the validity of the reported results.
- Model suitability for dataset: using a BiLSTM, a recurrent neural network architecture, for a tabular dataset with no temporal features is methodologically questionable.
- Comparison methods: the authors discuss comparisons with other models in the Results section but do not provide sufficient detail on the experimental setup, hyperparameter tuning, or metrics for each baseline.

Suggestions for Improvement:

Integrate the feature selection process into the cross-validation pipeline to ensure unbiased evaluation.
Provide a detailed justification for using BiLSTM+CNN on a dataset without temporal characteristics. Consider revising the model choice or explicitly discussing its benefits for this context.
Add a dedicated section in the Methods for baseline models, detailing hyperparameter configurations and the experimental protocol used.

Validity of the findings

Weaknesses:

-The choice of a hybrid model is not sufficiently justified given the dataset characteristics.
-The results may be overestimated due to potential data leakage in the feature selection process.
-The reproducibility of the work is hindered by the lack of detail on baseline setups.

Suggestions for Improvement:

-Include an ablation study focusing on the role of each component (BiLSTM and CNN) and justify their inclusion.
-Reassess the results using a validation process that integrates feature selection within the cross-validation folds.
-Provide the code or a detailed algorithm for replicating the experiments, including baseline comparisons.

---

## Round 0.2 · accepted · Accept

I accept the paper! It was well improved!

Reviewer 1 ·

Basic reporting

The authors have addressed all concerns regarding basic reporting:
- Clarity & Readability: They performed a thorough proofreading and revised complex sentences to enhance readability.
- Literature Review: They incorporated more recent studies on XAI in healthcare, including 2024 references.
- Structure & Visuals: Figures (especially SHAP visualizations) were improved, and captions were clarified.
- Reproducibility & Data Sharing: They uploaded their code to GitHub, enhancing transparency.
- Definitions & Terminology: A glossary was added, and key technical terms (e.g., SHAP, BiLSTM) were clearly defined.
Given these comprehensive revisions, I find the basic reporting satisfactory and accept the authors' revisions.

Experimental design

The authors have sufficiently addressed all concerns regarding the experimental design:
1. Methodology Details:
- They provided additional details on model hyperparameters, training configurations, and the SHAP implementation.
- Two new subsections were added: “Comprehensive Design of the BiLSTM-CNN Model” and “SHAP Integration for Explainable Breast Cancer Diagnosis” to enhance clarity and reproducibility.
2. Justification for CNN Use in Tabular Data:
- They included a clear explanation of why CNN was used alongside BiLSTM and how it was adapted for tabular datasets.
3. Dataset Limitations & Generalizability:
- The authors explicitly acknowledged the limitations of using the Wisconsin Breast Cancer Diagnostic dataset and its impact on generalizability.
- They incorporated a cross-validation strategy and discussed the need for external validation with diverse datasets in "Dataset Diversity and External Validation" under Conclusions and Future Work.
4. Explainability Techniques:
- The rationale for selecting SHAP over LIME and permutation importance was clearly explained in the “Explainable AI (XAI) with SHAP” section.
5. Reproducibility & Code Availability:
- The authors uploaded their code to GitHub, making their work transparent and reproducible.
Given these revisions, I find the experimental design sufficiently improved and accept the authors' revisions.

Validity of the findings

The authors have adequately addressed concerns regarding the validity of their findings:
- Novelty & Impact: They clarified the unique contributions of the BiLSTM-CNN model and justified its application to tabular data.
- Data Robustness & Statistical Validity: They acknowledged dataset limitations, added cross-validation details, and included statistical validation techniques (confidence intervals, error analysis).
- Class Imbalance: They provided a detailed discussion on data balancing and its impact on performance.
- Reproducibility & Transparency: Code is now available on GitHub for validation.
- Conclusions: Revised to temper claims about clinical applicability and emphasize future validation.
These revisions sufficiently improve the paper, and I accept the authors' response.

Additional comments

The authors have made significant improvements based on the feedback provided. The manuscript is now clearer, more rigorous, and better structured. Key revisions include:
- Improved clarity and readability through proofreading.
- Expanded literature review with recent studies.
- Enhanced methodology details, including model design, CNN justification, and statistical validation.
- Increased transparency, with code now available on GitHub.
- Refined conclusions, acknowledging dataset limitations and future research directions.
These changes effectively strengthen the study, and I accept the revisions.

Reviewer 4 ·

Basic reporting

The manuscript presents a well-structured study on breast cancer detection using a hybrid deep learning approach with Explainable AI (XAI). The topic is relevant, addressing the critical need for interpretable AI in medical diagnostics.
• Language and Clarity
The manuscript is well-written, but there are some areas where clarity can be improved. Certain sections contain grammatical inconsistencies.
• Literature Review and Background
The manuscript provides a comprehensive background, citing relevant works to support the research context. However, the discussion on prior works could be expanded to further highlight the specific contributions of this study compared to existing methods. The research gap is stated, but a clearer articulation of how this work advances the field beyond previous studies would strengthen the motivation.
Recommendation: Minor revisions to refine language, clarify the research gap, and improve figure presentation.

Experimental design

The study presents a well-defined methodology and employs rigorous experimental design principles. The integration of Explainable AI (SHAP) with a BiLSTM-CNN model is an interesting approach to improving both accuracy and interpretability in breast cancer classification.
• Research Question and Scope
The research question is well-defined and aligns with the journal’s scope. However, the rationale for selecting SHAP over other XAI methods (e.g., LIME, Grad-CAM) could be more explicitly justified. A comparative discussion on different interpretability techniques would add value.
• Methodological Rigor
Additional details on hyperparameter selection and tuning should be included to improve transparency and facilitate replication.
Recommendation: Minor revisions to provide further justification for SHAP, expand on hyperparameter tuning details, and acknowledge dataset limitations.

Validity of the findings

The study presents strong performance results, with the proposed BiLSTM-CNN model achieving high classification accuracy. However, some areas require further scrutiny to ensure the validity and robustness of the findings.
• Statistical Soundness and Overfitting Concerns
The reported accuracy (99.3%) and AUC-ROC (1.00) indicate exceptionally high performance. While this suggests a highly effective model, it also raises concerns about potential overfitting. It would be beneficial to test the model on an external dataset to confirm its generalizability. A discussion on how overfitting was mitigated, beyond the use of dropout layers, should also be included.
Recommendation: Major revision required to address potential overfitting, and improve external.

Additional comments

No comments